# Parent’s Perception of the Types of Support Given to Families with an Infant with Phenylketonuria

**DOI:** 10.3390/nu15102328

**Published:** 2023-05-16

**Authors:** Sophie Cassidy, Sharon Evans, Alex Pinto, Anne Daly, Catherine Ashmore, Suzanne Ford, Sharon Buckley, Anita MacDonald

**Affiliations:** 1School of Health Sciences, Faculty of Health, Education and Life Sciences, City South Campus, Birmingham City University, Birmingham B15 3TN, UK; sophie.cassidy@mail.bcu.ac.uk; 2Birmingham Women’s and Children’s NHS Foundation Trust, Birmingham B4 6NH, UK; evanss21@me.com (S.E.); alex.pinto@nhs.net (A.P.); a.daly3@nhs.net (A.D.); catherine.ashmore@nhs.net (C.A.); 3NSPKU National Society for Phenylketonuria, Sheffield S12 9ET, UK; suzanne.ford@nspku.org; 4North Bristol NHS Trust, Bristol BS10 5NB, UK; 5Department of Psychology, Faculty of Health, Psychology and Social Care, Manchester Campus, Manchester Metropolitan University, 53 Bonsall Street, Manchester M15 6GX, UK; sbuckley2@me.com

**Keywords:** PKU, phenylketonuria, social media, Facebook, healthcare professionals, dietitians, support

## Abstract

Background: A diagnosis of phenylketonuria (PKU) in an infant is a devastating and overwhelming event for their parents. Providing appropriate information and support is paramount, especially at the beginning of a child’s life. Investigating if parents are receiving the right support is important for continued care. Methodology: An online survey was distributed to explore parents’ perceptions of current support and information provided by their healthcare provider and to rate sources of other support (*n* = 169 participants). Results: Dietitians received the highest (85%) rate of “very helpful” support. Overall, parents found Facebook to be helpful for support but had mixed reactions when asked if healthcare professionals (HCPs) should provide advice as part of the groups. When rating the most effective learning methods, the top three were 1:1 teaching sessions (*n* = 109, 70%), picture books (*n* = 73, 50%), and written handouts (*n* = 70, 46%). Conclusion: Most parents are happy with the support and information they receive from their dietitian but required more support from other HCPs. Facebook groups provide parents with the social support that HCPs and their family may be unable to offer, suggesting a place for social media in future PKU care.

## 1. Introduction

Phenylketonuria (PKU) is a rare inherited disorder of amino acid metabolism that is caused by pathogenic variants in the phenylalanine hydroxylase (PAH) gene. PAH deficiency causes an elevation in blood phenylalanine and its accumulation in the brain and other tissues due to an inability to metabolise phenylalanine. It causes substantial intellectual disability, developmental delay, and behavioural issues if untreated [1]. PKU is detected via newborn screening, allowing early treatment that is highly effective in the prevention of severe neurocognitive impairments [2]. In the UK, infants are screened at 5 days of age, and it is recommended that treatment is initiated within the first 10 days of life in any infant with a blood phenylalanine >360 µmol/L. A newborn screening nurse typically communicates the initial positive newborn screening result to the parents. This is followed by an immediate appointment within 24 h at a specialist metabolic centre in order for treatment to commence without delay.

PKU is a life-altering condition, and although it can be controlled, it necessitates a challenging and relentless treatment. In childhood, management focuses on a reduction in blood phenylalanine through restrictions in dietary phenylalanine intake, with or without the addition of sapropterin. Parents are the primary caregivers, and ongoing management of PKU is associated with a high parental burden. Immediately, they must learn how to manage a phenylalanine-restricted diet and facilitate regular home sampling of blood spots for phenylalanine, and at the same time, they must deal with the emotional turmoil of a PKU diagnosis in their child. They must constantly live and cope with the medical, psychological, and social consequences of this condition [3]. Parents commonly feel isolated and lonely. They may experience depression and anxiety [4,5], although this may not occur in all parents of children with PKU [6,7,8].

A range of parental changes may be necessary following a diagnosis of PKU: a reduction in working hours, enabling a parent to stay at home to care for their child; modifications to lifestyle, particularly when associated with lower financial income; the need to secure extra support from family members; and the ongoing intrusive impact of treatment on life. The development of a collaborative relationship between families and their healthcare providers is essential.

Throughout a child’s early years, parents will receive health professional support, especially from specialist metabolic dietitians. In the UK, care is provided by a multidisciplinary team including metabolic physicians, dietitians, and nurses. Children generally attend a PKU clinic twice per year, whereby they see all team members, but dietitians are in contact with families more frequently than any other health professional discipline. Dietitians talk to the families at least weekly about the patient’s blood phenylalanine results, and they may review the patients at home, online, or at additional hospital clinics. They may visit schools or nurseries and liaise with the child’s primary care general practitioner (GP) about the prescription of special low-protein foods and protein substitutes [9,10]. Working together, all health professionals should deliver an extensive education programme and support parents as they navigate their way through the challenges of infant feeding, weaning, administration of protein substitutes, and preparation of low-protein foods and the everyday management at different childhood stages [11]. Education may be provided via one-to-one or group teaching sessions, material handouts, or supportive/information apps. Over the years, there has been inconsistent literature on how dietitians can best support parents. A study by Bernstein et al. [12] examined parent and patient perspectives of learning methods and found that one-to-one counselling was the most effective method. A further study showed that group sessions was the most effective method of learning [13].

In contrast to traditional methods of patient education, social media has become important in the transmission of knowledge and education. Social media is based on highly interactive, electronic platforms in which individuals and communities share, co-create, modify, and discuss user-generated content. It provides immediate and copious information. Facebook is the most popular social network globally, with 44% of the UK population using Facebook daily. The average user spends 23 min on Facebook each day, and the most active age group is 25 to 34 year olds in the UK [14]. Generally, mothers use Facebook as a parenting support network more than fathers do [15]. By contrast parents spend only a relatively short time with health professionals, so social media offers complementary support and a platform for parents to share their stories. However, there is no research examining the impact of social media platforms on PKU care and the reliability of shared information. Furthermore, with the changing landscape of how parents obtain medical information, dietitians and other professionals working with IMD may need to consider using social media as an educational method more often.

Any intervention that improves parental knowledge or child management strategies and supports parents through their early years of caring for their child with PKU will help to optimize care and minimize any parental stress and anxiety. To ensure the dietetic profession stays relevant in a rapidly changing environment, we need to continue to evaluate the best way to support patients [16]. The present study aims to explore parental perceptions about the care they receive following a diagnosis of PKU in their children and the role that social media plays in education and support systems.

## 2. Materials and Methods

A cross-sectional online survey was developed for UK parents/carers of children with PKU. The unvalidated questionnaire was designed using *Online Surveys* (https://www.onlinesurveys.ac.uk, accessed on 1 May 2021) by a master’s degree student (SC) with assistance from qualified dietitians experienced in research and PKU dietary management (AM and AP). The questionnaire was pilot tested by service users to check for understanding and appropriateness.

### 2.1. Questionnaire

The questionnaire was uploaded onto the UK National Society for PKU (*NSPKU*) website between 3 May and 4 September 2021, inviting parents/carers of children in the UK with PKU aged ≤12 years to respond. It was also promoted on the *NSPKU* Twitter account and *UK and Ireland’s Facebook Group*. The questionnaire (Appendix A) consisted of 6 demographic questions (3 yes/no and 3 multiple choice), and 30 topic-specific questions: 9 multiple choice, 8 Likert scale (5-point), 5 yes/no, and 8 open-ended. Questions addressed parent perspectives on the support and information they have received from healthcare professionals, family, friends, and social media regarding their child’s PKU, and the degree to which they trust the information provided. The questions also explored parental ideas for future support.

### 2.2. Data Analysis

The percentage of responses was calculated. A thematic analysis was used for open-ended questions to convert responses into common descriptions and patterns [17]. Verbatim quotes were included to support the reported quantitative data.

### 2.3. Ethical Approval

Ethical approval was obtained from Birmingham City University (Cassidy/#9172/sub2/r(B)/2021/Mar/HELS FAEC). Informed consent was obtained from all respondents via an online information sheet and consent form. If they agreed with all the online statements in the consent form, they selected a box to declare their consent to participate.

## 3. Results

### 3.1. Demographic Data

There were 169 UK respondents, representing approximately 20% of all children with PKU in the UK aged 12 years or younger [9]. One hundred and fifty four (92%) were mothers and fourteen (8%) were fathers of children with PKU aged ≤12 years. Thirteen percent (*n* = 22) were single parents, and sixteen percent (*n* = 27) did not speak English as their first language. Fifty-two percent (*n* = 88) of respondents had only one child in the household, forty-one percent (*n* = 69) had two or three children, and seven percent (*n* = 12) had four or more children.

Other than the respondent, *n* = 149 (88%) reported other people caring for their child with PKU, including 67% (*n* = 100) grandparents, 24% (*n* = 35) other relatives, 32% (*n* = 47) nurseries, 6% (*n* = 9) older siblings, 4% (*n* = 6) friends/neighbours, 5% (*n* = 8) after school clubs, and 3% (*n* = 5) childminders.

### 3.2. Initial Feelings on Diagnosis of PKU

Following a diagnosis of PKU in their child, parents reported a range of feelings (Table 1). Open-ended comments described emotions such as devastated, guilty, worried, scared, nervous, lost, distressed, broken, suicidal, shocked, upset, frightened, heartbroken, gutted, and “*I cried a lot*”. For some parents, these feelings persisted over time: “*I still feel the same way 5 years on*”. Parents with more than one child with PKU found second or subsequent diagnoses less stressful, reporting feelings of peace, acceptance, neutrality, and “*a fierce sense of being ready to do whatever was necessary to make sure that PKU didn’t win*”. For them, PKU became “*our normal*”, and one family of three children reported that their non-PKU child felt like “*the odd one out*”.

### 3.3. Support from and Trust in Healthcare Professionals

When asked to rank their support from various HCPs in the early years, from 1 (not very helpful) to 5 (very helpful), PKU dietitians and nurses were the most helpful, whilst GPs and health visitors were the least helpful (Table 2). Parents were particularly satisfied with dietetic information and support (79%, *n* = 133), with comments such as “grateful”, “reassured”, “confident”, and “informed”. A small number reported anxiety (4%, *n* = 7), frustration (3%, *n* = 5), or disappointment (2%, *n* = 4) with their dietitian.

Most parents (78%) reported no support from a psychologist. Three parents reported that when there were other medical issues not associated with PKU (e.g., diabetes and autism), support was fragmented, leading to feelings of isolation and mistrust in HCPs.

When asked how much they trusted the information on PKU provided by HCPs, the responses were similar to the results for the level of support that was provided by HCPs (Table 2).

Parents described their need for consistent and emotional support from their PKU team and expressed their frustrations with community HCPs, who were involved post diagnosis of PKU but not part of the core PKU team. Participant comments included:“*It is absolutely crucial to have a consistent person to deal with when learning about PKU. Someone who actually knows your family and the child with PKU.*”“*GPs don’t know what PKU is. I feel I have to inform them when I see them.*”“*The support from the health visitor did not extend to making any effort to understand PKU or what that might mean. There appeared to be no attempt to liaise with the hospital.*”“*There was no emotional support and no follow up at all regarding my own well-being.*”

### 3.4. Where Parents Go for Support and Information

When asked who outside the hospital team they found most helpful for support, listening to their concerns and making them feel better, just over half of respondents said family, the *NSPKU*, and parents of other children with PKU (Table 3). In the free-text section for adding other types of support not listed, 12% (*n* = 21) said their dietitian and 6% (*n* = 10) Facebook groups. Comments included: “*Dietitian offers a lot of support, mentally and emotionally*” and “*The most supportive place I have found is the Facebook PKU group. There are always people to answer questions and lots of great food ideas are shared*”.

Half of respondents had never attended a NSPKU conference, but of those who had, 41% (*n* = 29/71) found them helpful or very helpful: ‘*the conference was really useful as this was the first time I was able to meet other families and started to feel less isolated*.’

Not including HCPs, the following were listed as the most helpful education resources: information booklets (74%), the NSPKU website (68%), other parents of children with PKU (54%), and hospital PKU events (53%) such as cooking demonstrations (Table 3): “*Cooking demonstrations were really helpful and gave me confidence with the [low protein] products, which I needed despite being a competent cook with ‘regular’ food*”.

Family and friends were ranked as the least helpful in providing information (56% and 63%), with comments indicating that obtaining support from family and friends was difficult “*as they struggle to understand and comprehend the diet and the obstacles that come with it*”.

### 3.5. Use of Social Media for Support and Information

Eighty-six percent (*n* = 144) of respondents used social media for PKU information. Facebook groups were the most used (75%, *n* = 125) for both support and information (Table 4) and were also considered the most helpful by respondents, followed by Twitter. Among social media users, over 40 different PKU Facebook groups were listed by respondents; *PKU UK and Ireland* was the most commonly used (67%, *n* = 72/107), followed by the *NSPKU* group (19%, *n* = 20), *Support for PKU parents* (8%, *n* = 9), *PKU world wide support group* (6%, *n* = 6), and *PKU Friendly* (5%, *n* = 5). Forty-one percent (*n* = 44/107) of parents had joined more than one PKU Facebook group.

Three quarters (*n* = 95/126) of the respondents trusted or somewhat trusted the information provided by Facebook groups and half (*n* = 62/125) said they relied on Facebook groups for support. Comments included: “*If not sure about what I see, I ask the dietitian for clarification*” and “*always check against other sources*”. The benefits that respondents identified from being in a PKU Facebook group were divided into five main themes from the 109 responses (Table 5). They particularly valued the support and a sense of belonging to a community of people in a similar situation; the meal, product, or recipe ideas; and the quick access to hints, tips, and advice.

Nearly half (47%, *n* = 78/167) of respondents thought that HCPs should be more involved in the social media groups, 8% (*n* = 13) disagreed, and 43% (*n* = 71) were unsure (Table 6).

### 3.6. Preferred Method of Obtaining Dietary Information

Respondents identified that the most useful method of obtaining dietary information from their dietitian was one-to-one training sessions either in person, by telephone, or remotely, followed by written information and picture booklets (Figure 1). However, all listed methods were popular with 49% or more of the parents.

When learning about PKU, 66% (*n* = 111) of parents said that they like to have the dietitian there to support them; 29% (*n* = 49) prefer to obtain some support and information from the dietitian and then conduct their own research; and 4% (*n* = 6) prefer to learn independently and require minimal dietetic support. Participant comments included:“*I prefer to do things through my dietitian. We feel safe this way.*”“*Early on I preferred to have the dietitian there throughout the process, but with time I do feel more confident to do my own research.*”

### 3.7. Early Education Received

At the start of treatment, 40% (*n* = 66/167) of parents reported that their child was breast-fed in addition to a phenylalanine-free amino acid infant formula, and 59% (*n* = 98/167) used a combination of standard and phenylalanine-free infant formula. In total, 55 (33%) respondents reported on the support they received with breastfeeding: 71% (*n* = 39) received advice from a midwife, 16% (*n* = 9) accessed voluntary breastfeeding support groups, and 6% (*n* = 3) the national breastfeeding helpline. Only 29% (*n* = 16) were loaned a breast pump and 18% (*n* = 10) received training on how to use the pump.

Participant comments included:“*I think it was assumed that because I was already breastfeeding, the best thing would be to continue. However we couldn’t get the hang of switching between bottle and breast.*”“*The early feeding support was handled by midwives and breastfeeding advisors but I would have preferred more input from people who knew the specific challenges of early PKU feeding.*”“*More training for dietitians on [breast] feeding would have been beneficial as I believe they are in the best place to advise and support from the beginning.*”

Dietetic advice on important topics were generally received by most parents (Figure 2).

Most parents (>80%) received information on taking heel prick blood spots (Table 7). Information on dental hygiene, vitamin supplementation, and child development was received by less than 50% of parents.

## 4. Discussion

The results of this survey in a group of parents of children with PKU highlighted the communication, education, and support that the parents of an infant with PKU received following diagnosis from health professionals, their family, and the wider PKU community through social media (particularly Facebook). In the UK, infants with PKU are diagnosed and managed within a standard framework, but there were still differences in the way parents were informed about PKU and supported. New parents of children with PKU have several emotional, practical, information, social, and psychological needs, but not all of these were met by the healthcare professional support package they were offered. PKU places a high burden on parental care. Inadequate communication strategies and support adversely affect parental emotional outcomes, and even many months after the diagnosis, parents describe episodes of depression and despair.

The skills and attributes of the person communicating the result was a crucial factor in the appropriate of newborn screening results, as giving bad news is always challenging. The results of this survey indicated that parents could remember what was said and how it was communicated. Some parents were told not to seek information from the internet about PKU but then said they sought information from this resource almost immediately. Parents said they commonly did not receive enough immediate information or reassurance about PKU to help allay their fears until their first hospital appointment, which may have been delayed until the following day. In the UK, although the initial screening results are usually given by a newborn screening nurse, parents reported that it was commonly delivered by telephone. Using this method for giving distressing news is a particularly difficult way of communication as at this stage, no healthcare professional relationship has been established with the parents; it is harder to express empathy as this cannot be demonstrated by body language or facial expressions. It is impossible for the healthcare professionals to assess the reaction or level of distress of the parents they are talking with. It is also difficult to ensure that both parents are present to support each other and to talk to them both at the same time [18].

Parents commonly described the initial diagnosis period as a traumatic time, requiring considerable emotional adjustment. Parents were overwhelmed with sadness, disbelief, guilt, and self-blame, displaying many similarities to a grief response. Some parents described the false reassurance given by the midwife when the heel prick blood spot was taken for screening. Some felt overwhelmed with the diagnosis or had difficulty retaining the information that was given. These findings were similar to other studies that show that receiving a positive newborn screening result leave parents anxious, stressed, angry, and distressed [19,20,21].

Creating a trusting environment for parents is foundational [22]. Positive and continual interactions with health professionals are important in establishing effective patient–professional relationships and acceptance of the condition. Poor communication can contribute to a rejection of PKU or dissatisfaction with care. Most respondents in this study reported that they trusted and valued the information given by their dietitian above other professionals, and 85% were very happy with their support. Even with social media, most parents prefer their dietitian to be present throughout the process of caring for their child. The survey results indicated that the support from dietitians extended beyond clinical matters, commonly acting as an advocate for their child and establishing long-term collaborative partnerships with parents. Parents were less happy with the service when there was no consistent dietitian or when the dietitian was inexperienced.

In contrast, parents were not happy with the provision of care by health visitors, midwives, and GPs. Not only did these professionals have limited or no experience with PKU, parents also felt that some were disinterested in gaining new knowledge. They were concerned that this may have a negative impact on both the quality and their access to care, particularly as in the UK, GPs prescribed protein substitutes and special low protein foods for PKU treatment. Parents had limited psychological support, even though many parents felt that they had struggled with their mental health during the diagnosis period and it is known that psychological barriers can impact on parental ability to manage dietary management for their children [23]. Lord and colleagues [24] examined parents’ reactions to PKU and showed that a significant proportion of parents (5% of fathers and 12% of mothers) reported clinical levels of posttraumatic reactions, even years after the diagnosis. Many UK PKU clinics do not have a psychology service that would have the resources to provide regular support to new families.

Family and social relationships should be important to parents in coping with a child with a chronic condition and appear to play a role in moderating reactions to stress associated with the condition [25]. In our survey, over 20% of parents regarded their family support as either unhelpful or very unhelpful. Parents were frustrated at repeatedly having to explain their child’s condition to social contacts. e.g., nursery and school teachers, organisers of children's activities, and restaurant staff. Parents valued attending group educational events but noted that the provision of these varied throughout the country. They considered that support from other parents of children with PKU, the *NSPKU*, and conferences was important, although less than half of respondents had the opportunity to attend the *NSPKU* annual conference.

Given the lack of support from some HCPs and family, it is not surprising that many parents of children with PKU turn to social media for reassurance. This study highlighted the role of social media groups within the PKU community as a method enabling parents to connect with each other and to share experiences during challenging periods, with over 65% of parents finding Facebook support groups helpful. Facebook posts from and about people living with PKU made the condition more visible and less intimidating. It enabled parents to express their emotions about daily life and coping with PKU. The majority used the online social groups for recipes and ideas from people in the PKU community, hopefully, enabling them to adhere to treatment plans and to improve self-care practices. Experienced parents provided support to parents of newly diagnosed infants. Some parents used Facebook by regularly posting messages or uploading photos, while others were passive users, checking others’ pages, pictures, and updates only [26,27]. In non-PKU conditions, it has been shown that having social support from others who have a shared experience can help relieve stress and provide a shared social identity [20,28]. They provide practical and emotional support, a shared empathic experience, that they did not receive from their families and friends [29,30]. One study found that more than half of parents would rather discuss the condition with peers than a professional [31]. Facebook groups may not cater to all, but there is limited evidence of these groups negatively affecting parents’ ability to care for their child with PKU [32,33]. A systematic review of 42 studies on social media suggested that only 7% of studies were associated with patient harm [34]. In some cases, there may be an over reliance on Facebook, with comparisons and judgments of each other, privacy concerns, and misleading information. PKU Facebook groups also consist of a highly heterogeneous membership, with people with different treatment experiences, which may cause some confusion, but Facebook moderators can help provide guidance.

Facebook groups may also offer healthcare professionals in PKU an opportunity to give both individuals and group support, advice, and encouragement to foster self-management and behaviour change [35]. The rapid transfer of information, low cost, and broad availability of social media make it an attractive platform for managing care, communication, and interventions in chronic disorders [34]. Some of the participants in this survey suggested that HCPs should be available on Facebook to monitor the accuracy of the information given and to correct any misinformation. However, there are several barriers to this. Healthcare institution policies on the use of social media by healthcare staff need to be flexible to allow this type of communication to occur [35]. Furthermore, some respondents considered that social media was a place for expressing their opinions and emotions and that it should be separate from any discussion with professionals. Interestingly, some even considered that the health professionals may become the victims of abuse, as Facebook does provide a platform for antagonistic cultures and norms [36].

In PKU, engaging patients and their families in education is important as they require comprehensive knowledge about how to care for their children. They require nutritional health literacy skills, effective parenting skills, and resources beyond those normally required by parents in general. The most popular learning methods were 1:1 teaching sessions and handouts (written and picture books), with group sessions further down the ranking. Parents valued the technique of education ‘chunking’, i.e., providing the information in small chunks over time, giving them time to assimilate new information [37]. The results from this survey contrast some of the evidence base, with one paper reporting that 78% of participants (patients with PKU and parents) found handouts to be the least effective method of information provision and 91% of parents saying they would attend a group clinic if offered [12]. Group sessions were not considered a popular choice for education in the current study. Although parents received comprehensive dietary information, they reported needing more help on the practicalities of breast feeding; knowledge about using breast milk pumps; and education on dental hygiene, use of vitamin supplements, and child development and expected milestones.

This study has several limitations. This questionnaire was not validated. Likert scales could lead to central tendency bias, with participants avoiding the ‘extreme’ options, as well as acquiescence bias, as participants were aware that this was a dietetic study, leading to further potential bias when discussing dietitians [38]. As participants were only provided with a description of the extremes on the scale, the in-between responses were subject to interpretation [39]. The respondents were not randomly selected, and participation was voluntary. Additionally, individuals without internet access may have been unable to participate. The survey was promoted on the NSPKU Twitter account and on the UK and Ireland Facebook group, meaning participants were more likely to be informed and proactive about PKU because of their engagement on social media platforms. Therefore, the survey population may not be representative of the entire PKU population.

## 5. Conclusions

In summary, providing appropriate information and support to parents is paramount when an infant is diagnosed with PKU. The present findings highlight parental preferences for the type of support and learning styles. Most parents were happy with the support and information they receive from their dietitian but require more support from other HCPs. It is important that health professionals recognize the importance of social media as a vehicle for support, information, and communication for families of children with PKU. Facebook groups provide parents with the social support that HCPs and their family may be unable to offer, suggesting a place for social media in future PKU care. The role of HCPs in the provision of information about PKU on social media should be explored further.

## Figures and Tables

**Figure 1 nutrients-15-02328-f001:**
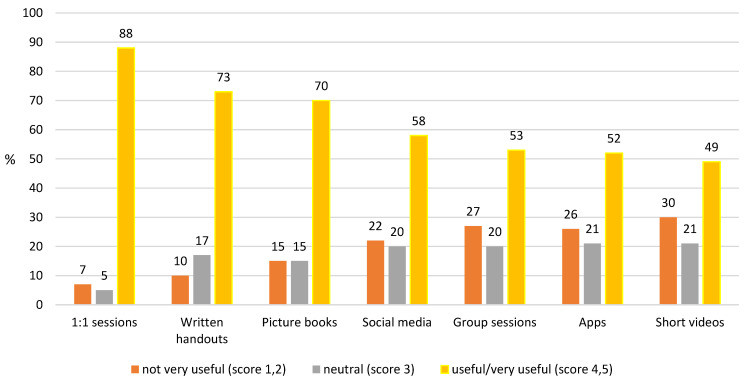
Preferred method of obtaining diet information.

**Figure 2 nutrients-15-02328-f002:**
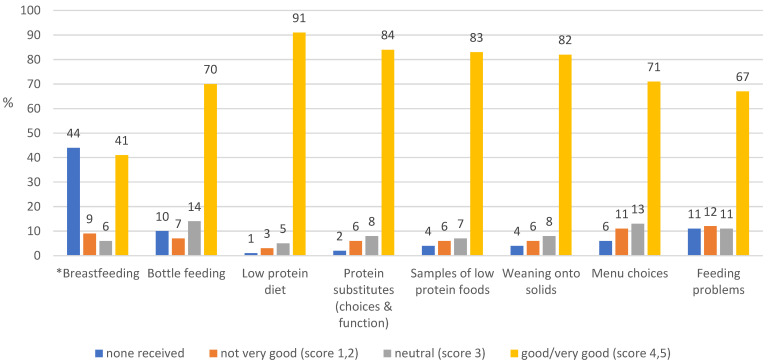
Ranking of advice received from dietitians during the first year of diagnosis (%) (* note: breastfeeding and choice of feeding method were usually established by the time of PKU diagnosis).

**Table 1 nutrients-15-02328-t001:** Parental feeling following child diagnosis of PKU.

Parental Feelings/Emotions on PKU Diagnosis	% (*n*)		% (*n*)
Sad	83 (141)	Lonely	38 (64)
Anxious	81 (136)	Angry	36 (60)
Overwhelmed	81 (136)	Neutral	3 (5)
Confused	57 (97)	Calm	2 (3)
Exhausted	40 (68)	Relaxed	<1 (1)
**Common thematic open-ended responses**			
**Fear of the unknown/lack of knowledge**			15 (26)
“*It took a lot of time to understand the condition and to fully comprehend the care needed in order to keep [her] safe.*” “*Scared of the unknown. What would the future be like for my child.*” “*I remember thinking I wish my child had another disease that she could be cured of as she was never going to be free of PKU. I thought she was going to have a very poor quality of life.*” “*We were told to get to the hospital immediately. We hadn’t a clue what was happening.*”			
**Overwhelming**			8 (13)
“*Never got over it—felt been hit by a train. It was heart breaking, I still cry.*” “*Felt like a massive life changing event and left to deal with it.*” “*When my child was approaching 1 year I had an emotional breakdown coming to terms with his diagnosis and all that it entailed for me.*”			
**Shock**			5 (9)
“*[He] is my 7th child and my only one with PKU -it came as a complete shock.*” “*It was a shock and caused us a lot of worry especially as an older cousin in the family has PKU with learning difficulties.*”			
**How the diagnosis was conveyed**			5 (8)
“*I was given the diagnosis over the phone out of hours and told to be at the hospital first thing the next morning, this led to a sleepless night as we were provided with minimal information and no support or guidance.*” “*We were told face to face in our house and I think this is the best way for this to be done.*” “*The midwife told us it could lead to brain damage and learning difficulties, those words stuck in mine and my partners mind when we didn’t know any other information about it or the fact that he will be fine on a low protein diet.*” “*I was called by a nurse who gave us the shocking and life-changing diagnosis and told us not to Google it but to look at the information she would email us. It’s unrealistic to tell people not to look it up on the internet.*”			
**Diagnosis with knowledge of PKU, i.e., sibling or relative**			5 (8)
“*1st baby (PKU); Absolutely broken. World had come crashing down around me. 2nd & 3rd babies (1 PKU, 1 not); Peace. Acceptance. Intense love. We all deal with what it is.*” “*We were already aware of PKU due to another family member and that its manageable by diet so we didn’t have the initial fears that parents that haven’t heard of PKU would have.*”			
**Guilt**			4 (7)
“*Intense guilt for having wanted my son so badly and yet giving him this burden of a life (as I saw it then).*” “*Thinking that it was caused by us as parents.*”			
**Isolation**			3 (5)
“*I felt hugely isolated and lonely in the 1st 5 years.*”			
**Resignation/acceptance**			3 (5)
“*Me and the husband said it is what it is, having PKU doesn’t affect quality of life, our daughter will still develop normally. We can either cry about it or just get on with it, so we’ve decided to just get on with it and support our daughter in any way we can.*”			

**Table 2 nutrients-15-02328-t002:** Support from and trust in HCPs in the early years of a PKU diagnosis.

	Very Helpful/Helpful (Score 4, 5)	Neutral (Score 3)	Not Very/ Not Helpful (Score 1, 2)	Not Applicable
**How helpful was the support from different HCPs in the early years of a PKU diagnosis (%)**				
Dietitian	92	4	4	1
Nurse	64	17	9	11
Hospital doctor	50	22	21	7
Health visitor	22	17	58	3
GP	12	16	66	7
Psychologist	5	3	14	78
**How much do you trust the information provided by healthcare professionals (%)**				
Dietitian	94	4	1	1
Nurse	73	12	3	12
Hospital doctor	61	16	16	7
Health visitor	22	15	58	6
GP	15	17	59	9
Psychologist	9	4	13	74

**Table 3 nutrients-15-02328-t003:** Where people go for support and how much they trust in the information provided.

	Very Helpful/Helpful (Score 4, 5)	Neutral (Score 3)	Not Very/ Not Helpful (Score 1, 2)	Not Applicable
**Where do you go to for support, and who listens to your concerns and makes you feel better (%)**				
Family	53	22	21	3
Other PKU parents	53	12	13	21
NSPKU	52	10	21	17
Friends	28	30	36	6
NSPKU conferences	19	10	19	52
**Where do you obtain your PKU information from (%)**				
Information books	74	16	8	2
NSPKU	68	7	17	9
Other PKU parents	54	16	15	14
Hospital PKU events	53	6	11	30
Company learning packages	43	16	24	17
NSPKU conferences	27	3	21	49
Family	12	15	56	17
Friends	9	7	63	21

**Table 4 nutrients-15-02328-t004:** How helpful people find social media at providing support and information for PKU.

	Very Helpful/Helpful (Score 4, 5)	Neutral (Score 3)	Not Very/ Not Helpful (Score 1, 2)	Not Applicable
**How helpful are social media sites at providing support for PKU (%)**				
Facebook	68	13	10	9
Twitter	52	8	9	30
Instagram	27	11	19	43
YouTube	16	16	18	50
Internet Forums	7	2	24	67
**How helpful are social media sites at providing information for PKU (%)**				
Facebook	69	11	12	7
Twitter	59	3	9	29
Instagram	25	11	19	45
YouTube	19	11	20	50
Internet Forums	7	2	27	65

**Table 5 nutrients-15-02328-t005:** Respondent benefits from being in a Facebook group.

Themes	% (*n*)
**Support/understanding and sense of community/belonging**	51 (56)
“*Offloading, emotional support.*” “*Solidarity and understanding from a community of people who “get it” in a way others don’t.*” “*Being able to empathise with other parents who are going/have gone through similar things as well as being able to help other parents feel part of a PKU family where we share ideas and ask for advice.*” “*It’s good to know we’re not alone and we can laugh, cry or moan about PKU related stuff.*”	
**Quick access to hints, tips, advice, solutions to problems, and potential ideas**	40 (44)
“*Answers and ideas from people who live with PKU day in, day out.*” “*Quick and helpful responses of things I’m not sure about and need to know quickly.*” “*Don’t even have to engage with people as can just search for things people have added in the past.*” “*Good for ideas. But for proper information I go to my dietitian.*”	
**Food/meal/recipe ideas**	40 (44)
“*Loads of recipe ideas and new product finds.*” “*I’m able to ask about certain foods which if I didn’t have this group I would be constantly bothering the dietitians!*”	
**Reassurance**	10 (11)
“*Reassurance that things will be okay.*” “*To hear other parents struggling with similar issues we have so we know it’s normal to have bad days.*” “*Comfort in knowing I’m not the only parent going through this.*”	
**Negative experiences**	7 (8)
“*Don’t ask for advice as often there are mixed opinions and often turns quite negative*” “*Interesting to see other views, but it’s often conflicting and can be quite argumentative if people disagree with your approach.*” “*Some people who post I don’t trust their calculations of exchanges.*” “*There’s a lot of misinformation out there.*” “*I don’t trust what is said. I only trust my dietitian. She knows what she is talking about.*”	

**Table 6 nutrients-15-02328-t006:** Pros and cons to healthcare professionals being more involved in social media groups.

Themes	% (*n* = 104)
**Pros**	
**For expertise and advice:**	32 (35)
“*Some more professional advice online could be good as PKU is very complicated and not all nutritional information on food packages is easy to understand.*” “*Twitter is good because there are dietitians on there who give great support and advice.*”	
**To correct inaccurate information:**	12 (12)
“*Sometimes what has been said may not be correct so if they see wrong information being shared, would be useful for them to set the record straight.*” “*This would give me more confidence in the knowledge we receive and stop a lot of conflictive responses.*”	
**To learn about PKU from a parent/patient perspective:**	9 (9)
“*I feel like they would gain a lot more knowledge just from being in the groups and looking at the information others provide and the struggles we all face as parents with children who have PKU or as adults who have PKU.*” “*They would have more of an understanding of what our worries are and what sort of questions we ask.*”	
**Cons**	
**People may feel less inclined to share/it is for sharing opinions:**	10 (10)
“*People might not open up as much to professionals as they do to other parents, therefore they get more out of it if professionals stay away.*” “*Sometimes parents want to write about a particular difficulty they having with a healthcare professional and look to other parents for advice in how to deal with that.*”	
**Not their role/too busy/no time:**	10 (10)
“*I don’t think health care professionals would have time to become involved in social media groups.*”	
**Input from different hospitals/centres:**	6 (6)
“*Too vast to control the quantity of people’s ideas and challenge behaviour especially when they are led by varying trusts and dietitians with different principles of care.*”	
**Professional vulnerability**	4 (4)
“*I feel that health care professionals would be vulnerable to the abuse associated with a lot of social media.*”	
**Other options**	
**Do not recommend relying on Facebook for facts or would go back to dietitian**	7 (7)
“*I would always go back for official advice from our dietitians.*” “*I wouldn’t trust the majority of the information on social media as some of it is coming from people who are off diet and think there doing fantastic and some is from people who have no clue. But then I see posts about some people unable to see their consultant or dietitian to ask questions so they turn to Facebook for advice.*”	
**Separate groups for HCP and parents or specific times for HCPs to join in**	3 (3)
“*Perhaps have a drop in slot where they can answer generic questions* i.e., *on exchange values or suitable medication.*” “*It would be helpful if there was a dedicated time on social media where parents could chat about concerns in a more relaxed way, sometimes I don’t want to ring up to the hospital as it feels like I’m being a pain.*”	

**Table 7 nutrients-15-02328-t007:** Advice from HCPs during the early days.

Advice Given	% (*n*) Receiving Advice
How to take a blood test	93 (156) *
When to take a blood test	82 (137) *
Immunisations	59 (99) **
Child development, e.g., speech/language	38 (63) **
Vitamin supplements	30 (50) **
Dental hygiene	25 (42) **

* 167 respondents; ** 168 respondents.

## Data Availability

Not applicable.

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
