# Peer review of "Parent’s Perception of the Types of Support Given to Families with an Infant with Phenylketonuria"

_nutrients, 2023, doi:10.3390/nu15102328_

Round 1

Reviewer 1 Report

Hi

Thank you for your paper, I enjoyed reading it. It was well written and easy to follow. You explored a few important and previously under researched  topics-in particular social medias role in parents experience of rare diseases and who parents find the whole experience, especially the first few weeks of having a child with a rare disease.  

As I am not from the UK there were are few points that I would like clarification on in order the appreciate the paper more. How many  PKU families with children less than 12yrs are there in the UK? thus what is your denominator-what percentage of these families responded.  Throughout the paper there is a lack of any over denominator (eg in those that attend PKU conferences ) and thus its impossible to draw too many conclusions

You note that 44% of adults in the uk are on Facebook and yet I would have thought this would be much higher in younger child bearing adults? I am trying to get an idea if you had a good representative same size or not and couldn't get this from the paper.  Also who is part of the 'team' the sees the PKU neonate/infant? Where I am from that is a metabolic consultant , nurse specialist and dietician and I suspect, well I hope (!) , that the doctor and nurse would get more of a favourable response if the study was carried out here.  Do the metabolic paediatricians not see the PKU patients?  While clearly the dieticians are always key could you include a brief sentence or two on who the UK system works-i suspect it is quite varied depending on location.  Can dieticians not prescribe PKU products in the UK and is this why it is left up to the GP's?  

The paper is very prone to ascertainment bias-those with an agenda, those who access social media , those with problems etc are more likely to respond -you do note this briefly at the end but I felt the paper was reported a somewhat negative element of the PKU family community.  Perhaps im wrong but my personal, and extensive experience, is that the families are generally very appreciative and supportive of the metabolic teams support.  

Author Response

Thank you for your paper, I enjoyed reading it. It was well written and easy to follow. You explored a few important and previously under researched  topics-in particular social medias role in parents experience of rare diseases and how parents find the whole experience, especially the first few weeks of having a child with a rare disease.  

  • Thank you for the positive feedback.

As I am not from the UK there were are few points that I would like clarification on in order the appreciate the paper more. How many  PKU families with children less than 12yrs are there in the UK? thus what is your denominator-what percentage of these families responded.  Throughout the paper there is a lack of any over denominator (eg in those that attend PKU conferences ) and thus its impossible to draw too many conclusions

  • We have added a sentence under the methods to say that the respondents represent approximately 20% of children with PKU in the UK aged 12 years and younger. In the UK, around 75 infants with PKU are diagnosed each year and commence dietary treatment. The National Society for PKU has one annual conference. It is expensive and run in different places around the UK and so may not be accessible to all. Around 60% of attendees are adolescents and 40% children with PKU. Only around 100 patients attend this annual conference so only a small percentage of children under 12 years of age.

You note that 44% of adults in the UK are on Facebook and yet I would have thought this would be much higher in younger child bearing adults? I am trying to get an idea if you had a good representative same size or not and couldn't get this from the paper.  

  • Thank you. We state that 44% of the UK population (not PKU population) use Facebook daily. This includes young and old people but more of the population will be registered to a Facebook site but not use it daily. In our survey, 86% of parents of children with PKU used social media. This is most of our respondents.

Also who is part of the 'team' the sees the PKU neonate/infant? Where I am from that is a metabolic consultant, nurse specialist and dietician and I suspect, well I hope (!) , that the doctor and nurse would get more of a favourable response if the study was carried out here. Do the metabolic paediatricians not see the PKU patients?  While clearly the dieticians are always key could you include a brief sentence or two on who the UK system works-i suspect it is quite varied depending on location. Can dieticians not prescribe PKU products in the UK and is this why it is left up to the GP's?  

  • In centres in the UK, patients would be seen by a medical consultant, specialist nurse and metabolic dietitian. A psychologist is rarely available. The doctor and nurse see the patients twice per year in clinic only. There is commonly more than one metabolic paediatrician in each metabolic team – so families commonly see a different doctor when they attend clinic so are unable to build up a close relationship. However, dietitians are in contact more frequently than any other health professional. They talk to the families at least weekly with blood results, they see the patients at home or will have frequent online teaching sessions. They visit schools, nurseries, have contact at family meetings. They commonly act as a patient advocate and support with social issues, so dietetic input is intense and far outweighs the contribution of other health professionals. We have added text in the introduction to clarify the role of health professionals in the UK.
  • Dietitians do not prescribe protein substitutes and low protein foods in the community. The NHS budget for these products is within the community so prescriptions are done by GPs on the advice of dietitians. GPs do not receive any specific training in PKU, so they are not best placed to perform this role, and the managers of local health authorities are concerned about cost savings, so they frequently question parents on the need for low protein products and this leads to significant caregiver negativity toward GP practices. A sentence to clarify the role of GPs in the UK has also been added to the introduction.

The paper is very prone to ascertainment bias-those with an agenda, those who access social media , those with problems etc are more likely to respond -you do note this briefly at the end but I felt the paper was reported a somewhat negative element of the PKU family community.  Perhaps im wrong but my personal, and extensive experience, is that the families are generally very appreciative and supportive of the metabolic teams support.  

  • Thank you but there was no agenda to this paper. We reported the facts as we found them. We were surprised about the negative element from the PKU community, but this is what they reported.

Reviewer 2 Report

This paper presents results from an online survey used unvalidated questionnnaire about parents' perceptions, whose children live with PKU. The manuscript is well written, introduction is sufficient.

Methods: It is recommended that the questionnaire questions be made available upon request.

The presentation of the results is more descriptive using somehow too many individual wording in tables. Here I would recommend a shorten of the individual examples from responses. 

In table 2,3 and 4, percents summary are occasionally over 100%, please check it.

Discussion and conclusion are well written. 

Author Response

This paper presents results from an online survey used unvalidated questionnaire about parents' perceptions, whose children live with PKU. The manuscript is well written, introduction is sufficient.

  • Thank you for your positive feedback.

Methods: It is recommended that the questionnaire questions be made available upon request.

  • Thank you. The questionnaire questions have been included as supplementary data.

The presentation of the results is more descriptive using somehow too many individual wording in tables. Here I would recommend a shorten of the individual examples from responses. 

  • We acknowledge that there may be too many quotes, consequently we have deleted some of these as suggested.

In table 2,3 and 4, percents summary are occasionally over 100%, please check it.

  • Thank you for identifying these errors. We have checked all figures and made corrections where necessary. Some add to 101% due to rounding of individual figures.

Discussion and conclusion are well written. 

  • Thank you for your positive feedback.